# Alteration of *adeS* Contributes to Tigecycline Resistance and Collateral Sensitivity to Sulbactam in *Acinetobacter baumannii*

Yunxing Yang,[a] Xiaochen Liu,[b,c,d] Danyan Zhou,[e] Jintao He,[b,c,d] Qiong Chen,[a] Qingye Xu,[a] Shenghai Wu,[a] Weiying Zhang,[a] Yue Yao,[b,c,d] Ying Fu,[b,c,d] [ID] Xiaoting Hua,[b,c,d] [ID] Yunsong Yu,[b,c,d] [ID] Xianjun Wang[a]

[a]Department of Clinical Laboratory, Affiliated Hangzhou First People's Hospital, Zhejiang University School of Medicine, Hangzhou, China
[b]Department of Infectious Diseases, Sir Run Run Shaw Hospital, Zhejiang University School of Medicine, Hangzhou, Zhejiang, China
[c]Key Laboratory of Microbial Technology and Bioinformatics of Zhejiang Province, Zhejiang Institute of Microbiology, Hangzhou, Zhejiang, China
[d]Regional Medical Center for National Institute of Respiratory Diseases, Sir Run Run Shaw Hospital, School of Medicine, Zhejiang University, Hangzhou, Zhejiang, China
[e]Department of Clinical Laboratory, Xiangshan First People's Hospital Medical and Health Group, Ningbo, Zhejiang, China

Yunxing Yang, Xiaochen Liu, and Danyan Zhou contributed equally to this work and share first authorship. The order was determined by the corresponding author after negotiation.

**ABSTRACT** The treatment of extensively drug-resistant (XDR) *A. baumannii* has emerged as a major problem. Tigecycline (TGC) and sulbactam (SUL) are both effective antibiotics against XDR *A. baumannii*. Here, we investigated the in-host evolution and mechanism of collateral sensitivity (CS) phenomenon in development of tigecycline resistance accompanied by a concomitant increase of sulbactam susceptibility. A total of four XDR *A. baumannii* strains were sequentially isolated from the same patient suffering from bacteremia. Core-genome multilocus sequence typing separated all the strains into two clusters. Comparative analysis of isolate pair 1 revealed that multiplication of $bla_{OXA-23}$ within Tn*2006* on the chromosome contributed to the change in the antimicrobial susceptibility phenotype of isolate pair 1. Additionally, we observed the emergence of CS to sulbactam in isolate pair 2, as demonstrated by an 8-fold increase in the TGC MIC with a simultaneous 4-fold decrease in the SUL MIC. Compared to the parental strain Ab-3557, YZM-0406 showed partial deletion in the two-component system sensor *adeS*. Reconstruction of the *adeS* mutant in Ab-3557 *in situ* suggested that TGC resistance and CS to SUL were mainly caused by the mutation of *adeS*. Overall, our study reported a novel CS combination of TGC and SUL in *A. baumannii* and further revealed a mechanism of CS attributed to the mutation of *adeS*. This study provides a valuable foundation for developing effective regimens and sequential combinations of tigecycline and sulbactam against XDR *A. baumannii*.

**IMPORTANCE** Collateral sensitivity (CS) has become an increasingly common evolutionary trade-off during adaptive bacterial evolution. Here, we report a novel combination of tigecycline (TGC) resistance and CS to sulbactam (SUL) in *A. baumannii*. TGC and SUL are both effective antibiotics against XDR *A. baumannii*, and it is essential to reveal the mechanism of CS between TGC and SUL. In our study, the partial deletion of *adeS*, a two-component system sensor, was confirmed to be the key factor contributing to this CS phenomenon. This study provides a valuable foundation for developing effective regimens and sequential combinations of tigecycline and sulbactam against XDR *A. baumannii*.

**KEYWORDS** *Acinetobacter baumannii*, *adeS*, collateral sensitivity, tigecycline, sulbactam, seesaw effect

Address correspondence to Yunsong Yu, yvys119@zju.edu.cn, or Xianjun Wang, wangxianjun0913@163.com.

The authors declare no conflict of interest.

*A*cinetobacter baumannii is a ubiquitous and notorious pathogen and can cause various infections in hospital and community settings, such as pneumonia, bacteremia, urinary tract infection, and meningitis (1, 2). *A. baumannii* is a member of the ESKAPE (*Enterococcus faecium*, *Staphylococcus aureus*, *Klebsiella pneumoniae*, *Acinetobacter*

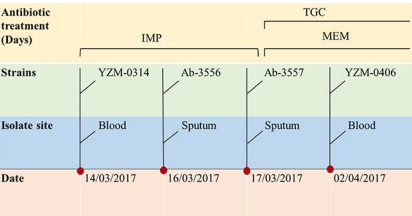

**FIG 1** Timeline of infection and antibiotic usage. IMP, imipenem; TGC, tigecycline; MEM, meropenem.

*baumannii, Pseudomonas aeruginosa, and Enterobacter* spp.) family of pathogens, which are well known for causing common and hard-to-treat infections (3). Extensively drug-resistant (XDR) *A. baumannii* has emerged as a major public health threat (4, 5).

Tigecycline (TGC) and sulbactam (SUL) are both good alternatives for treating XDR *A. baumannii* infections (6–8). Tigecycline is a glycylcycline antibiotic with a wide antibacterial spectrum that binds to the 30S subunit of the ribosome and inhibits amino acid synthesis (9). The extensive use of tigecycline has led to the evolution and spread of many resistance mechanisms (10). Prominent among these mechanisms is the overexpression of the resistance-nodulation-cell division (RND) efflux pump AdeABC, which is upregulated by the two-component system AdeRS (11). Other acquired mechanisms, such as mutation of *trm* and *plsC* and acquisition of *tetX3*, *tetX4*, and *tetA* variants, have also been studied (12–15). Sulbactam is a β-lactamase inhibitor that is also an effective antimicrobial agent against XDR *A. baumannii*. Sulbactam resistance is mainly conferred by a class of β-lactamases, such as TEM-1, ADC-30, and OXA-23 (16, 17). Resistance to SUL can also be attributed to alteration of the penicillin-binding protein 3 (18). Although the mechanisms underlying TGC and SUL resistance are diverse, the dynamic evolution of TGC and SUL resistance and the related mechanisms remain to be fully elucidated.

Under antimicrobial and host pressure, bacteria often make evolutionary trade-offs to adapt to the host environment. During the evolution process *in vivo*, the resistance mechanism against a specific antimicrobial class frequently confers increased sensitivity to other antimicrobial classes, a phenomenon termed collateral sensitivity (CS), also called "the seesaw effect" (19, 20). Antibiotic cycling or combination therapy based on CS antibiotic pairs may thus enable the efficient treatment of resistant strains (21, 22). CS associated with TGC resistance in *A. baumannii* is very rare. Interestingly, we observed the emergence of CS to SUL in *A. baumannii* after exposure to TGC. The aim of this study was to explore the evolutionary progress of TGC resistance and CS to sulbactam in *A. baumannii* and provide a foundation for formulating more effective treatment strategies against XDR *A. baumannii* based on CS.

## RESULTS

**Characteristics of *A. baumannii* strains from the same patient.** A total of four *A. baumannii* strains were isolated from sputum and blood of the same patient during hospitalization. The patient was admitted for acute exacerbation of chronic obstructive pulmonary disease, respiratory failure, and bloodstream infection. During hospitalization, imipenem (0.5 g every 8 h), meropenem (1.0 g every 8 h), and tigecycline (50 mg every 12 h) were administered successively. The timeline of *A. baumannii* infection and antibiotic usage is illustrated in Fig. 1. Strain YZM-0314 was first isolated from a blood sample, then Ab-3556 and Ab-3557 were collected from sputum consecutively. Another *A. baumannii* isolate, YZM-0406, was cultured from blood again after 2 weeks.

Whole-genome sequencing revealed four clinical isolates belonging to two sequence types (STs), ST547 and ST195. Based on the core-genome multilocus sequence typing (cgMLST) scheme, the selected strains were divided into two clusters. Strains Ab-3556 and YZM-0314 were grouped into the same cluster, while the two remaining strains (Ab-3557 and YZM-0406) were grouped into the other cluster. The number of allelic differences between the two clusters reached as high as 44, which suggested that the two clusters were from different origins.

**TABLE 1** Antimicrobial susceptibility of isolates used in the study[a]

| Strain | MIC (mg/liter) of antibiotic determined by: | | | | | | | | | | | | | |
| | Microbroth dilution | | | | Etest | | | | | | | | | |
| | SUL | IMP | CO | TGC | TZ | MC | CPS | LE | PM | MP | AK | TC | CI | PP |
|---|---|---|---|---|---|---|---|---|---|---|---|---|---|---|
| Isolate pair 1 | | | | | | | | | | | | | | |
| YZM-0314 | 32 | 16 | 0.5 | 4 | 64 | 16 | 32 | ≥32 | ≥256 | ≥32 | ≥256 | ≥256 | ≥32 | ≥256 |
| Ab-3556 | 128 | 32 | 0.5 | 4 | ≥256 | 8 | 64 | ≥32 | ≥256 | ≥32 | ≥256 | ≥256 | ≥32 | ≥256 |
| Isolate pair 2 | | | | | | | | | | | | | | |
| Ab-3557 | 64 | 16 | 0.5 | 1 | ≥256 | 8 | 192 | 16 | 192 | ≥32 | ≥256 | ≥256 | ≥32 | ≥256 |
| YZM-0406 | 16 | 16 | 0.5 | 8 | ≥256 | 32 | 32 | ≥32 | ≥256 | ≥32 | ≥256 | ≥256 | ≥32 | ≥256 |
| Mutant strain | | | | | | | | | | | | | | |
| Ab-3557-*adeS*-del 95-157 | 16 | 16 | 0.5 | 8 | ≥256 | 32 | 64 | ≥32 | ≥256 | ≥32 | ≥256 | ≥256 | ≥32 | ≥256 |
| QC strain | | | | | | | | | | | | | | |
| ATCC 25922 | 64 | 0.25 | 0.25 | 0.125 | 0.5 | 1.5 | 0.25 | 0.016 | 0.094 | 0.032 | 2 | 2 | 0.004 | 4 |

[a]YZM-0314, Ab-3556, Ab-3557, and YZM-0406 are clinical strains, and Ab-3557-adeS-del 95-157 is the mutant strain that was constructed *in situ*. SUL, sulbactam; IMP, imipenem; CO, colistin; TGC, tigecycline; TZ, ceftazidime; MC, minocycline; CPS, cefoperazone-sulbactam; LE, levofloxacin; PM, cefepime; MP, meropenem; AK, amikacin; TC, tetracycline; CI, ciprofloxacin; PP, piperacillin-tazobactam; QC, quality control.

The profile of the following resistance genes was similar: the $\beta$-lactam resistance genes $bla_{OXA-66}$, $bla_{OXA-23}$, and $bla_{ADC-73}$, the macrolide resistance genes *mph*(E) and *msr*(E), the tetracycline resistance gene *tet*(B), and the aminoglycoside resistance genes *strA*, *strB*, and *armA*. In addition, cluster 2 (Ab-3557 and YZM-0406) carried the $\beta$-lactam resistance gene $bla_{TEM-1D}$, the sulfonamide resistance gene *sul1*, and the aminoglycoside resistance genes *aac(6′)lb-cr*, *aadA24*, and *aph(3′)-la*.

**Change in antimicrobial susceptibility phenotype.** The antimicrobial susceptibility phenotype of all isolates was XDR, as they were resistant to cephalosporins, $\beta$-lactamase inhibitors, quinolones, carbapenems, and aminoglycosides, but not colistin (Table 1). The specific changes in the antimicrobial susceptibilities of the two isolate pairs, based on cgMLST, are listed below.

The resistomes of isolate pair 1 (YZM-0314 and Ab-3556) were similar, and phenotypic changes were observed for sulbactam, ceftazidime, and imipenem. Ab-3556 exhibited decreased susceptibility to sulbactam (4-fold) compared to YZM-0314, and a similar trend was observed for the MICs of ceftazidime and imipenem.

A significant difference in the AST profile of isolate pair 2 (Ab-3557 and YZM-0406) was observed for sulbactam and tigecycline. The MIC of tigecycline for YZM-0406 (8 mg/liter) was 8-fold higher than that for Ab-3557 (1 mg/liter). Additionally, the MICs of minocycline, levofloxacin, and cefepime for YZM-0406 were also elevated. The MIC of sulbactam for YZM-0406 (16 mg/liter) was 4-fold lower than that for Ab-3557 (64 mg/liter), and the MIC of cefoperazone-sulbactam for YZM-0406 was also decreased.

**Comparative genomic analysis of the two isolate pairs.** To uncover the mechanism of the changes in the AST profile, comparative genomic analysis of the isogenic pairs of isolates was performed. Comparison of the whole-genome sequence of YZM-0314 and Ab-3556 revealed that Ab-3556 carried two copies of $bla_{OXA-23}$ within Tn*2006* transposons, while YZM-0314 possessed only one copy of $bla_{OXA-23}$ (Fig. 2). The expression of $bla_{OXA-23}$ was determined by semiquantitative reverse transcription-PCR (qRT-PCR), and the mRNA expression of $bla_{OXA-23}$ in Ab-3556 was statistically higher than that in YZM-0314 ($P = 0.028$) (Fig. 3a). Multiplication of $bla_{OXA-23}$ could explain the increase of the MICs of imipenem, ceftazidime, and sulbactam.

For isolate pair 2, the deletion of bp 95 to 157 of *adeS* was detected in strain YZM-0406 (Fig. 4a), which resulted in the alteration of most amino acids of *adeS*, as shown in Fig. 4b. The mutation of *adeS* is a common cause of tigecycline resistance in *A. baumannii* and also could regulate many genes related to antimicrobial resistance and virulence, but the association with sulbactam susceptibility restoration remains unknown.

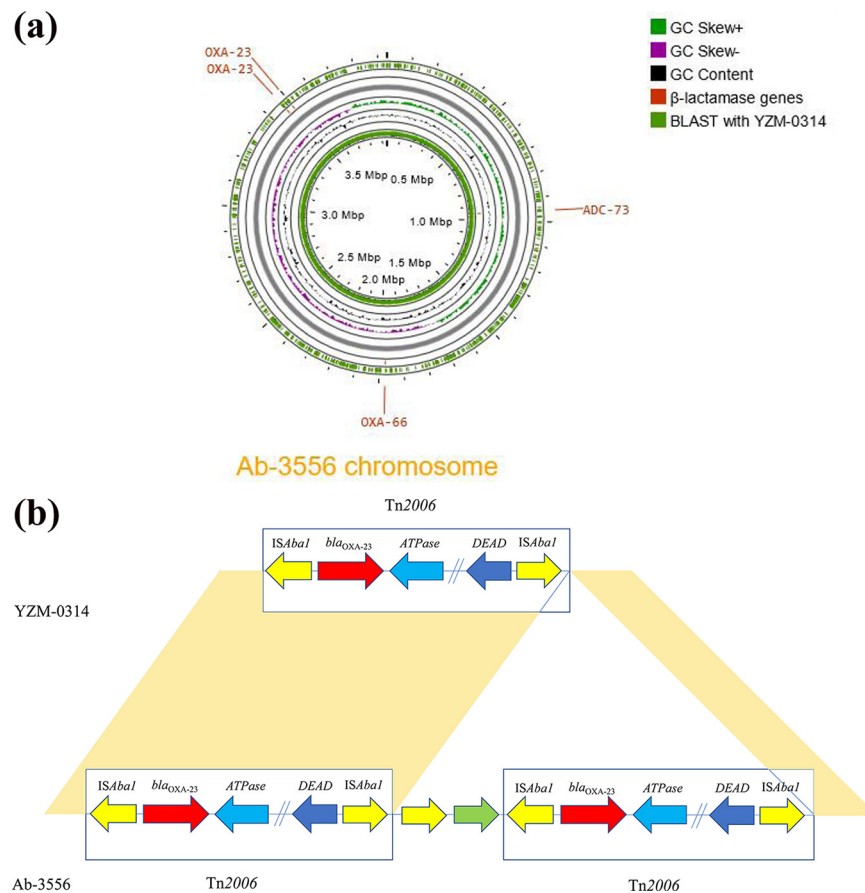

**FIG 2** (a) Genome map of the chromosome of Ab-3556. The word in red represents *β*-lactamase genes. Functions are color coded according to the key. The two peak circles represent the G+C content (black circle) and GC skew information (green and purple circles). (b) Schematic structure of the *bla*~OXA-23~-containing transposase in strains YZM-0314 and Ab-3556. The yellow shaded areas are the same in the two strains. The rectangle fragment is Tn*2006*, which was duplicated in Ab-3556.

**Identification of the *adeS* mutant associated with restoring susceptibility to sulbactam.** To identify the effect of the *adeS*-Δ95-157 mutant in restoring the susceptibility to sulbactam, we introduced the *adeS* mutant into the genome of the parental strain Ab-3557 *in situ*. The mutant Ab-3557-*adeS*-Δ95-157 displayed an elevated MIC of tigecycline compared to Ab-3557, and the MICs of minocycline, levofloxacin, and cefepime were also increased (Table 1). The relative expression of the efflux pump gene *adeB* in strain Ab-3557-*adeS*-Δ95-157 was statistically higher than that in parental isolate Ab-3557 ($P = 0.017$), which verified the upregulation effect of the novel *adeS* mutant on the expression of efflux pump AdeABC (Fig. 3b).

Interestingly, in contrast to the change in the MIC of tigecycline, the MIC of sulbactam for Ab-3557-*adeS*-Δ95-157 decreased from 64 mg/liter to 16 mg/liter. This observation

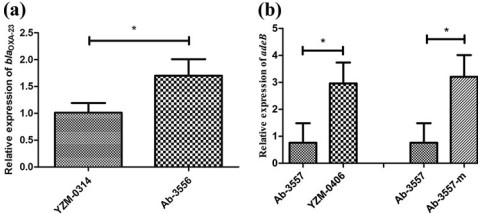

**FIG 3** Relative expression levels of the *bla*~OXA-23~ (a) and *adeB* (b) genes. YZM-0314 and Ab-3557 were used as the reference strains. The bars represent the means ± standard deviations of triplicate biology repeats. Mean differences of expression of *bla*~OXA-23~ and *adeB* were compared using the unpaired *t* test. *, $P < 0.05$.

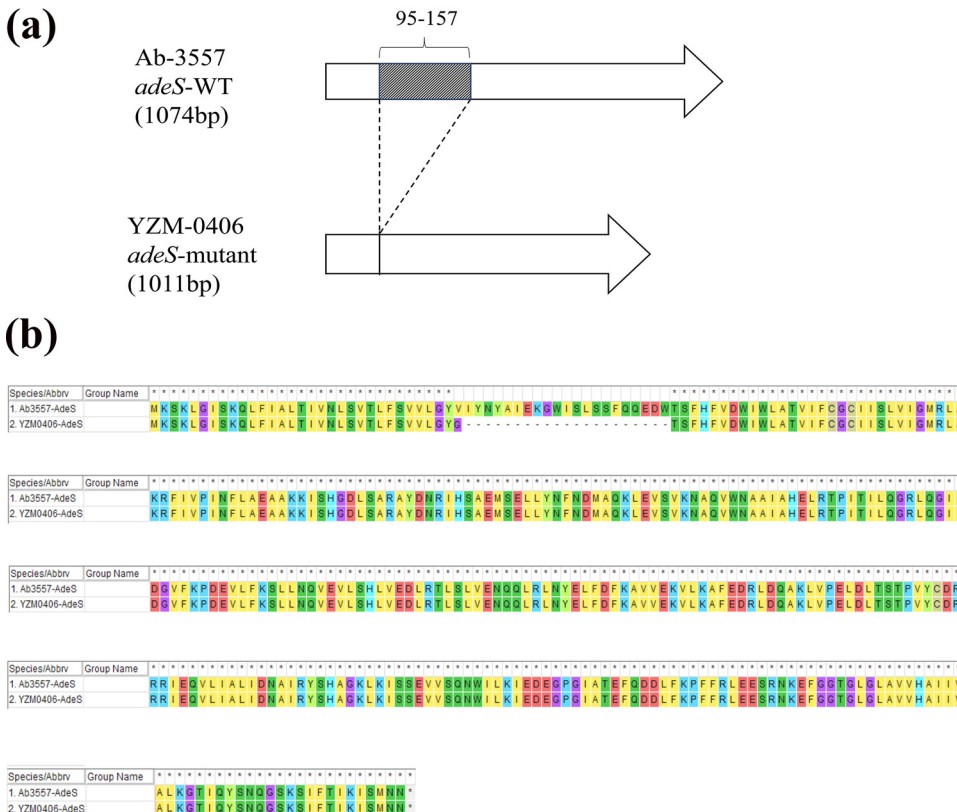

**FIG 4** Schematic diagram of the *adeS* gene (a) and amino acid sequence (b) comparing strains Ab-3557 and YZM-0406. The shaded rectangle is the deletion fragment of *adeS*.

demonstrated that the *adeS* mutant played a significant role in restoring sulbactam susceptibility.

## DISCUSSION

We report here a case of a patient infected with two isogenic isolate pairs of *A. baumannii* obtained from the blood and respiratory tract. Strain Ab-3556 of isolate pair 1 became less susceptible to sulbactam and carbapenem after approximately 1 week of carbapenem therapy. Comparative genomic analysis indicated that duplication of $bla_{OXA-23}$ contributed to increased OXA-23 expression and reduced susceptibility to sulbactam and carbapenem. The antimicrobial phenotype of isolate pair 2 presented an interesting phenomenon known as collateral sensitivity, i.e., the acquisition of tigecycline resistance simultaneously conferred increased sensitivity to sulbactam after prolonged tigecycline exposure. Alteration of the two-component system sensor *adeS* was verified as the key factor contributing to tigecycline resistance and collateral sensitivity to sulbactam.

Gene duplication and amplification (GDA) is common and directly associated with antibiotic resistance (23). The GDA of many resistance genes, such as $bla_{NDM-5}$, $bla_{TEM-1}$, and $bla_{OXA-23}$, has been reported to be associated with antimicrobial resistance (16, 24, 25). In this study, we identified a duplication of $bla_{OXA-23}$ within the Tn*2006* transposon during the evolution of carbapenem and sulbactam resistance *in vivo*. Multiplication of the carbapenemase gene $bla_{OXA-23}$ within Tn*2006* or Tn*2009* was also detected in the clinical strains and the serial passage induction strain *in vitro*, but the copy number of $bla_{OXA-23}$ did not correlate well with the MICs of imipenem and sulbactam due to the multifactorial resistance mechanism and different genomic background of the clinical strain (25, 26). In the isogenic sequential pairs of strains or the induced strain, amplification of $bla_{OXA-23}$ resulted in the development of cefoperazone-sulbactam and carbapenem resistance (27), which was similar to the results of our study.

Collateral sensitivity (CS) has become an increasingly common evolutionary trade-off during adaptive bacterial evolution (28). The CS of many antibiotic pairs, such as rifampin and tetracycline in *Enterococcus faecalis* and daptomycin and β-lactams in *Staphylococcus aureus*, may be widely found in many genera (29–32). On the basis of CS, some new treatment strategies for CS cycling or combination were used sequentially to treat infections and slow the development of antimicrobial agent resistance (21, 33). Here, we reported a novel antibiotic pair with tigecycline resistance and CS to sulbactam in *A. baumannii*. Resistance to tigecycline is easily induced *in vivo* and *in vitro*, so sensitization to sulbactam, which is effective against XDR *A. baumannii*, has important clinical implications. It is essential to uncover the underlying mechanism of tigecycline resistance with collateral sensitivity to sulbactam. In our study, a truncated *adeS*, a two-component system sensor, was confirmed to be the key factor contributing to tigecycline resistance and collateral sensitivity to sulbactam. As reported previously, the insertion of IS*Aba1* or gene alteration of *adeS*, leading to the upregulation of the RND efflux pump *adeABC*, was associated with tigecycline resistance (34). The deletion within *adeS* in our study, which was mainly located on the sensor domain (35), would result in the prevention of autophosphorylation of AdeS and inhibit the transfer of the phosphate group to AdeR. Thus, AdeR might be binding loosely to the intercistronic spacer region, leading to an overexpression of AdeABC (36). The upregulation of the efflux pump AdeABC could also explain the reduced susceptibility to tigecycline and other antimicrobial agents in our study. To our knowledge, the tigecycline resistance-related mutation of *adeS*-Δ95-157 in our study has not been described previously. The overexpression of the efflux pump *adeB* in the *adeS* mutant *in situ* also verified the effect of the novel mutant type of *adeS*. The mechanism of *adeS* mutation resulting in tigecycline resistance is fully understood; however, the relationship between the *adeS* mutation and increased sulbactam susceptibility is a hitherto-undescribed mechanism. *adeS*, a sensor of a two-component system, regulates many genes. Richmond et al. reported that deletion of the two-component system AdeRS modulated the expression of many genes, notably those related to antimicrobial resistance and virulence interactions (37, 38). However, the regulatory mechanism by which *adeS* affects the increased sensitivity to sulbactam is still poorly understood. Hence, the possible regulatory pathway of *adeS* mutation requires further study.

Resistance evolution for the persistence of one strain and new infection by strains of different genetic backgrounds often prolong hospital stays and lead to treatment failure. The identification of new infections caused by different strains and evolutionary research seem to be challenges to traditional microbiology, but whole-genome sequencing could provide the opportunity to address these challenges. Comparative genomic analysis of all the available *A. baumannii* strains from the same patient in our study has greatly facilitated the current understanding of the evolution of *A. baumannii in vivo* and simultaneously revealed the mechanism of collateral sensitivity. The in-depth investigation of the mechanism of collateral sensitivity and evolution would provide a valuable foundation for developing effective regimens and sequential combinations of sulbactam and tigecycline for use against XDR *A. baumannii*.

This study also had some limitations. The isolates were collected from only one clinical case, which indicate a lack of universality. Further studies will focus on collecting more clinical isolate pairs of *A. baumannii* to support the mechanism proposed. Moreover, the mechanism of increasing sensitivity to sulbactam mediated by the *adeS* mutant has not been studied in depth. Transcriptome and proteome research should be carried out further, and the regulatory networks and signal transduction pathways acted upon by *adeS* in *A. baumannii* should be given importance.

In summary, we have reported a case of infection caused by *A. baumannii* strains of different genetic background and tracked how XDR *A. baumannii* developed resistance under antibiotic exposure. The evolution of resistance to carbapenem and sulbactam can be attributed to the multiplication of *bla*$_{OXA-23}$ on the chromosome. Additionally, this is the first study reporting tigecycline resistance with CS to sulbactam *in vivo*, which is related to the mutation of the two-component system sensor *adeS*. These results explain the

molecular mechanism of resistance evolution and CS of TGC and SUL and lay a foundation for developing effective regimens and combination therapies against XDR *A. baumannii.*

## MATERIALS AND METHODS

**Strains and patient characteristics.** A total of four *A. baumannii* strains were sequentially isolated from the same patient in the Sir Run Shaw Hospital, School of Medicine, Zhejiang University, China, in 2017. These isolates were cultured from specimens obtained from sputum and blood. The strains were identified with a matrix-assisted laser desorption ionization–time of flight mass spectrometer (Bruker Daltonics, Billerica, MA, USA).

**Antimicrobial susceptibility testing.** The MICs of sulbactam, tigecycline, colistin, and imipenem were determined using the broth microdilution method according to the guidelines provided by the Clinical and Laboratory Standards Institute (CLSI) (39). The MICs of other commonly used antibiotics listed in Table 1 were determined using Etest strips (AB bioMérieux, France). The results for the antimicrobial agents tested were interpreted according to the CLSI breakpoints. *Escherichia coli* ATCC 25922 was used as the quality control strain.

**Whole-genome sequencing.** The genomic DNA of the strains in the study was extracted by using a QIAamp DNA minikit (Qiagen, Valencia, CA) according to the manufacturer's recommendations. The genome was sequenced on the Illumina HiSeq (Illumina, San Diego, CA, USA) and PacBio RS II (Pacific Biosciences, Menlo Park, CA) platforms. The reads were assembled by Canu and PacBio hierarchical genome assembly process workflows.

The assembled genomes were annotated using Prokka (https://proksee.ca), and antimicrobial resistance genes were detected using the ResFinder tool (https://cge.cbs.dtu.dk/services/ResFinder/). Comparative genome analyses were performed with breseq v0.33.0 (40). The putative mutant was confirmed by PCR and Sanger sequencing. The minimum-spanning tree was conducted based on cgMLST analysis with a core genome of 2,390 alleles using Ridom SeqSphere+ v7.2.3 software (Ridom GmbH, Münster, Germany) with default parameters (41).

**Reconstruction of the mutation of *adeS in situ*.** Reconstruction of the mutation of *adeS in situ* was performed by slightly modifying a method described in our previous study (42). Upstream and downstream sequences of the *adeS* mutant were amplified from YZM-0406, and the purified PCR products were cloned into the pMo130-Hyg$^R$ plasmid using the ClonExpress MultiS one-step cloning kit (Vazyme Biotech Co., Nanjing, China). The recombinant vector was introduced into the chromosome of Ab-3557 via electroporation and selected on Mueller-Hinton agar containing 100 mg/liter hygromycin. The clones that turned yellow after being sprayed with 0.45 mol/liter catechol were selected for PCR confirmation. The selected clones were cultured in MH broth with 10% sucrose overnight and then coated on MH agar. PCR and Sanger sequencing confirmed that the wild-type *adeS* in Ab-3557 was replaced by the *adeS* mutant.

**qRT-PCR of *bla*$_{OXA-23}$ and *adeB*.** The expression of *bla*$_{OXA-23}$ and efflux pump gene *adeB* was examined by qRT-PCR. RNA was extracted by using the Purelink RNA minikit with one-column DNase I digestion to remove any genomic DNA (Ambion, Carlsbad, CA, USA). The quantity, quality, and integrity of RNA were measured by an Agilent 2100 Bioanalyzer (Agilent Technologies, Waldbronn, Germany). Total RNA was reverse transcribed into cDNA using the PrimeScript RT reagent kit (TaKaRa, Dalian, China), and real-time quantitative RT-PCR was performed on an ABI7500 instrument (Applied Biosystems Inc., Foster, CA, USA). The primers used in this experiment are listed in Table S1 in the supplemental material. The housekeeping gene *rpoB* was used as an internal control. Finally, the expression of *bla*$_{OXA-23}$ and *adeB* in the strains was calculated by the threshold cycle ($2^{-\Delta\Delta CT}$) method (16). All experiments were performed in triplicate.

**Data analysis.** GraphPad Prism 5 software was used for graphical representation and statistical analysis. The change in the expression of *bla*$_{OXA-23}$ of strain Ab-3556 was calculated and compared with that of the reference strain YZM-0314, while differences in the mean expression level of *adeB* between the Ab-3557 control strain and other strains were assessed by an unpaired *t* test. A *P* value of <0.05 was considered statistically significant.

**Data availability.** The genomes of strains YZM-0314, Ab-3556, Ab-3557, and YZM-0406 have been deposited in the GenBank database under accession numbers CP104908 to CP104910, CP104785 to CP104787, CP104906 to CP104907, and CP104911 to CP104912, respectively.

## SUPPLEMENTAL MATERIAL

Supplemental material is available online only.
**SUPPLEMENTAL FILE 1**, PDF file, 0.06 MB.

## ACKNOWLEDGMENTS

This work was supported by grants from the National Key Research and Development Program of China (2018YFE0102100), the National Natural Science Foundation of China (82202561, 81861138054), and Natural Science Foundation of Zhejiang Province (LQ22H190005).

We report there are no competing interests to declare.

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
