## [Reviewer comments · Microbiology Spectrum]

Microbiology Spectrum

Alteration of *adeS* contributes to tigecycline resistance and collateral sensitivity to sulbactam in *Acinetobacter baumannii*

Yunxing Yang, Xiaochen Liu, Danyan Zhou, Jintao He, Qiong Chen, Qingye Xu, Shenghai Wu, Weiyang Zhang, Yue Yao, Ying Fu, Xiaoting Hua, Yunsong Yu, and Xianjun Wang

Corresponding Author(s): Yunsong Yu, Zhejiang Provincial People's Hospital, Xianjun Wang, Affiliated Hangzhou First People's Hospital, Zhejiang University School of Medicine

Review Timeline:

Submission Date:	November 10, 2022
Editorial Decision:	February 8, 2023
Revision Received:	April 11, 2023
Accepted:	April 13, 2023

Editor: Ayush Kumar

Reviewer(s): The reviewers have opted to remain anonymous.

Transaction Report:

DOI: <https://doi.org/10.1128/spectrum.04594-22>

February 8, 2023

Prof. Xianjun Wang
Affiliated Hangzhou First People's Hospital, Zhejiang University School of Medicine
Department of Clinical Laboratory
No.4 Xueshi Road
hangzhou
China

Re: Spectrum04594-22 (Alteration of *adeS* contributes to tigecycline resistance and collateral sensitivity to sulbactam in *Acinetobacter baumannii*)

Dear Prof. Xianjun Wang:

Thank you for submitting your manuscript to Microbiology Spectrum. Your manuscript has been reviewed by two experts and they have suggested several modifications to your work. When submitting the revised version of your paper, please provide (1) point-by-point responses to the issues raised by the reviewers as file type "Response to Reviewers," not in your cover letter, and (2) a PDF file that indicates the changes from the original submission (by highlighting or underlining the changes) as file type "Marked Up Manuscript - For Review Only". Please use this link to submit your revised manuscript - we strongly recommend that you submit your paper within the next 60 days or reach out to me. Detailed instructions on submitting your revised paper are below.

Link Not Available

Sincerely,

Ayush Kumar

Journals Department
Reviewer comments:

Reviewer #1 (Comments for the Author):

The manuscript "Alteration of *adeS* contributes to tigecycline resistance and collateral sensitivity to sulbactam in *Acinetobacter baumannii*" by Yunxing Yang et al., describes the in-host evolution of strains of *A. baumannii* and the generation of collateral sensitivity to sulbactam after treatment with tigecycline. The strategy of the study is interesting and the findings are novel. The methods and controls are fine, although some details are missing in the methods part.

The following are some comments/suggestions:

- Please briefly describe your methods in addition to the references you provide (such as in lane 300).
- How did you check the integrity of the RNA for qRT-PCR?
- Did you use PrimeScript RT Reagent Kit with gDNA Eraser (Perfect Real Time)? If not, please provide information for DNase treatment.
- Please add the reference strains you used for qRT-PCR in the methods part.
- I think a statistical analysis could be added to the data in Figure 4.
- In Figure 5a, please add some color or shading to the part of adeS that comes after region 95-157, as it gives the impression that it is a shorter version of the gene, rather than a completely different sequence (as in 5b).
- What is your theory about sulbactam susceptibility in the adeS mutant? What genes are known to be controlled by adeS in addition to adeABC? Are there any porins that might be up-regulated?
- Please change "Enterococcus" to "Enterococcus" (lane 70)

Reviewer #2 (Comments for the Author):

Please refer to attached world file

Staff Comments:

Preparing Revision Guidelines

Please return the manuscript within 60 days; if you cannot complete the modification within this time period, please contact me. If you do not wish to modify the manuscript and prefer to submit it to another journal, please notify me of your decision immediately so that the manuscript may be formally withdrawn from consideration by Microbiology Spectrum.

This study describes the properties of 4 multi-drug (XDR) resistant *Acinetobacter baumannii* isolates that were sequentially isolated from a patient who suffered from bacteremia and was subjected to sequential carbapenem (i.e. imipenem and meropenem) and tigecycline chemotherapy treatments. The strains were characterized through whole-genome sequencing and antimicrobial susceptibility testing. MLST categorized the 4 strains into two independent strain lineages. The strain of lineage 1, which was isolated later from the patient, gained a duplication of transposon Tn2006 carrying the β -lactamase resistance gene *oxa-23*, which may explain the observed changes in the antimicrobial susceptibility phenotypes reported in Table 1. The later isolate of strain lineage 2 gained a deletion within gene *adeS* encoding a sensor kinase that controls the *adeABC* multidrug efflux pump. This mutation may explain increased resistance to tigecycline in agreement with previous observations. Interestingly, the latter strain concomitantly acquired increased sensitivity to the β -lactamase inhibitor Sulbactam, which is known to also have antibacterial activity by itself towards *A. baumannii*. Thus, increased resistance to tigecycline through mutation of *adeS* appears to be associated with higher susceptibility to Sulbactam, which is a novel and interesting observation. The latter conclusion is also supported by an experiment in which the *adeS* mutation was genetically engineered in the parental strain, thereby conferring the identical tigecycline/Sulbactam resistance/susceptibility phenotype as observed in the clinically evolved strain.

However, there are some ambiguities in the manuscript that need to be addressed before the study can be published.

Major:

- Lines 116-120; Fig. 1: There is apparently a mistake in the timeline. If the total timeline was 14 days, i.e. 6 days IMP treatment followed by 17 days MEM treatment, we cannot have 30 days of TIG treatment overlapping with the former treatments.
- Lines 124-128: Is Figure 2 required? It does not show any information beyond the description provided in the text.
- Lines 138-140: As there is no sensitive reference strain included in Table 1, it is difficult for the reader to judge whether the criteria of an XDR strain are met. It would be helpful to include this information in Table 1.
- Lines 160-163; Fig. 3: Is Tn2006 correctly depicted in Fig 3B? I think the interrupting double prime should be located between the genes encoding the ATPase and DEAD box helicase. Moreover, the color code in Fig. 3A is not referenced in the text. Please explain.
- Lines 168-174; Fig. 5: Apparently, it escaped the attention of the authors that the isolated *adeS* allele carries an in-frame deletion, i.e. amino acids 32-53 are deleted, but all other amino acids are shared with the wild-type allele except for glycine residue at position 32. This is important for the interpretation of the activity of the encoded sensor kinase. Please correct the alignment in Fig. 5B accordingly. Is kinase or phosphatase activity of the kinase expected to be affected? How can constitutive activation of *adeABC* expression be explained by the deletion within *AdeS*. Is the response regulator known to autophosphorylate with acetyl-phosphate *in vivo*? The mutation in *adeS* detected in the current study needs to be re-evaluated by the authors.
- Lines 187-189: I think the MIC for Sulbactam decreased from 64 to 16 mg/ml.
- Lines 271-276: Sulbactam is an irreversible inhibitor of β -lactamases but also known to display antibacterial activity in *A. baumannii* by targeting PBPs directly (i.e., PBP1a/b and PBP3), enzymes required for cell wall synthesis. Is it possible that a constitutively active *AdeABC* efflux

pump may increase local concentration of Sulbactam in the exterior surrounding the cells, thereby leaving more of the drug that could subsequently inhibit the PBPs? Can this be tested?

Minor:

- Is mentioning the isolate names in the abstract essential information? Please simplify the abstract.
- Lines 139-132: Can the distribution of the resistance genes be designated identical, if there was a duplication of oxa-23 in one of the isolates?
- Line 194: Should read: "Strain Ab-3556 of the isolate pair 1 of the..." ?
- Line 198: "...reduced susceptibility to Sulbactam..."
- Line 228-229: "...novel antibiotic pair with tigecycline resistance and collateral..."
- Line 244-247: Please describe briefly what is known about the relationship between adeS mutations and tigecycline resistance as this is important information needed to evaluate the effects of the adeS in-frame deletion detected in the current study.
- Line 267: First SUL, subsequently TGC? Please specify.
- Lines 348-350: For convenience of the user, it would be advantageous to assign accession numbers to the corresponding strains. Currently, they are not in a particular order.
- Line 551: β -lactamase

Reviewer comments and Responses:

Reviewer #1 (Comments for the Author):

The manuscript "Alteration of adeS contributes to tigecycline resistance and collateral sensitivity to sulbactam in Acinetobacter baumannii" by Yunxing Yang et al., describes the in-host evolution of strains of A. baumannii and the generation of collateral sensitivity to sulbactam after treatment with tigecycline.

The strategy of the study is interesting and the findings are novel. The methods and controls are fine, although some details are missing in the methods part.

The following are some comments/suggestions:

COMMENT: *Please briefly describe your methods in addition to the references you provide (such as in lane 300).*

RESPONSE: The description of method has been added in lane 305-308.

COMMENT: *How did you check the integrity of the RNA for qRT-PCR?*

RESPONSE: The integrity of RNA was measured by Agilent 2100 Bioanalyzer Agilent Technologies (Waldbronn, Germany). And the description of method has been added in lane 346-348.

COMMENT: *Did you use PrimeScript RT Reagent Kit with gDNA Eraser (Perfect Real Time)? If not, please provide information for DNase treatment.*

RESPONSE: We didn't use PrimeScript RT Reagent Kit with gDNA Eraser (Perfect Real Time)we , but we performed an on-column DNase I digestion to remove any genomic DNA carryover during the RNA purification procedure with PureLink® DNase treatment, while the RNA is bound on the Spin Cartridge. This protocol includes the PureLink® DNase treatment, followed by steps to complete the washing and elution of RNA (lane 344-345).

COMMENT: *Please add the reference strains you used for qRT-PCR in the methods part.*

RESPONSE: We agree with this suggestion and the reference strains have been added in lane 359-362.

COMMENT: *I think a statistical analysis could be added to the data in Figure 4.*

RESPONSE: We thank the reviewer for pointing out this issue. We have applied unpaired *t*-test and revealed a significant difference in *bla*_{OXA-23} ($P=0.028$), *adeB* between Ab-3557 and YZM-0406 ($P=0.022$) and *adeB* between Ab-3557 and Ab-3557-m ($P=0.017$). Furthermore, some statistical analysis and descriptions have added in the Materials and methods part (lane 357-363), Result part (lane 161-162, lane 180-181) and Figure legends (lane 572-573).

COMMENT: *In Figure 5a, please add some color or shading to the part of adeS that comes after region 95-157, as it gives the impression that it is a shorter version of the gene, rather than a*

completely different sequence (as in 5b).

RESPONSE: We apologize for the inappropriate alignment of AdeS amino acid. Actually, the *adeS* mutant gene in YZM-0406 was a shorter version of the wide type *adeS* gene with alignment again. The AdeS protein carries an in-frame deletion, i.e. amino acids 32-53 are deleted, but all other amino acids are shared with the wild-type allele except for glycine residue at position 32. So we have corrected the alignment of AdeS amino acid sequence between Ab-3557 and YZM-0406 in Figure 4b.

COMMENT: *What is your theory about sulbactam susceptibility in the adeS mutant? What genes are known to be controlled by adeS in addition to adeABC? Are there any porins that might be up-regulated?*

RESPONSE: The two-component system AdeRS regulates many genes that contribute to biofilm formation, *A. baumannii* pathogenesis and virulence *in vivo* besides AdeABC efflux pump [1,2,3]. However, no studies have found that any porins are up-regulated by AdeRS. In our study, the transcriptomic analysis of Ab-3557-*adeS*-mutation revealed that the gene expression of Outer-membrane protein A (OmpA) was up-regulated, which may be also regulated by AdeRS. OmpA is β -barrel porin and also play a role in antimicrobial resistance and virulence in *A. bauamnii* [4,5]. OmpA is mainly distributed on the outer membrane and the outer membrane vesicle (OMV), so we speculated that the sulbactam susceptibility restoring might be mediated by the following mechanism: (a) Change the distribution of OmpA: increasing the distribution of OmpA on the outer membrane enlarges the channel of sulbactam into bacteria; (b) Decrease the secretion of OMV containing β -lactamase: because sulbactam is easily hydrolyzed by β -lactamses. The extracellular hydrolysis of sulbactam is reduced by decreasing the formation of OMV and the secretion of β -lactamases. The above hypothesis is currently being tested in a large number of experiments.

References:

1. Tomaras AP, Dorsey CW, Edelmann RE, Actis LA. Attachment to and biofilm formation on abiotic surfaces by *Acinetobacter baumannii*: involvement of a novel chaperone-usher pili assembly system. *Microbiology*. 2003;149(Pt 12):3473–84.
2. Gaddy JA, Arivett BA, McConnell MJ, Lopez-Rojas R, Pachon J, Actis LA. Role of acetobactin-mediated iron acquisition functions in the interaction of *Acinetobacter baumannii* strain ATCC 19606T with human lung epithelial cells, *Galleria mellonella* caterpillars, and mice. *Infect Immun*. 2012;80(3):1015–24.
3. Murray GL, Tsyganov K, Kostoulas XP, Bulach DM, Powell D, Creek DJ, et al. Global gene expression profile of *Acinetobacter baumannii* during bacteremia. *J Infect Dis*. 2017;215(suppl 1):S52–7.
4. Iyer R, Moussa SH, Durand-Reville TF, Tommasi R, Miller A. *Acinetobacter baumannii* OmpA Is a Selective Antibiotic Permeant Porin. *ACS infectious diseases* 2018, 4(3): 373-381.
5. Huang W, Meng L, Chen Y, Dong Z, Peng Q. Bacterial outer membrane vesicles as potential biological nanomaterials for antibacterial therapy. *Acta biomaterialia* 2022, 140: 102-115.

COMMENT: *Please change "Enterococcus" to "Enterococcus" (lane 70)*

RESPONSE: The word has been corrected (lane 67).

Reviewer #2 (Comments for the Author):

This study describes the properties of 4 multi-drug (XDR) resistant *Acinetobacter baumannii* isolates that were sequentially isolated from a patient who suffered from bacteremia and was subjected to sequential carbapenem (i.e. imipenem and meropenem) and tigecycline chemotherapy treatments. The strains were characterized through whole-genome sequencing and antimicrobial susceptibility testing. MLST categorized the 4 strains into two independent strain lineages. The strain of lineage 1, which was isolated later from the patient, gained a duplication of transposon Tn2006 carrying the β -lactamase resistance gene *oxa-23*, which may explain the observed changes in the antimicrobial susceptibility phenotypes reported in Table 1. The later isolate of strain lineage 2 gained a deletion within gene *adeS* encoding a sensor kinase that controls the *adeABC* multidrug efflux pump. This mutation may explain increased resistance to tigecycline in agreement with previous observations. Interestingly, the latter strain concomitantly acquired increased sensitivity to the β -lactamase inhibitor Sulbactam, which is known to also have antibacterial activity by itself towards *A. baumannii*. Thus, increased resistance to tigecycline through mutation of *adeS* appears to be associated with higher susceptibility to Sulbactam, which is a novel and interesting observation. The latter conclusion is also supported by an experiment in which the *adeS* mutation was genetically engineered in the parental strain, thereby conferring the identical tigecycline/Sulbactam resistance/susceptibility phenotype as observed in the clinically evolved strain.

However, there are some ambiguities in the manuscript that need to be addressed before the study can be published.

Major Comment:

COMMENT: Lines 116-120; Fig. 1: There is apparently a mistake in the timeline. If the total timeline was 14 days, i.e. 6 days IMP treatment followed by 17 days MEM treatment, we cannot have 30 days of TIG treatment overlapping with the former treatments.

RESPONSE: We agree with the criticism. The total timeline of strain isolation was 14 days, but the timeline of drug usage is not only 14 days. So we have deleted the numbers in the parentheses and revised the right bracket to avoid misunderstanding (Fig 1).

Fig 1. Timeline of infection and antibiotic usage. IMP, imipenem; TGC, tigecycline; MEM, meropenem; The number in parentheses is the number of days of medication.

COMMENT: *Lines 124-128: Is Figure 2 required? It does not show any information beyond the description provided in the text.*

RESPONSE: We agree with your suggestion, and we have deleted the Figure 2.

COMMENT: *Lines 138-140: As there is no sensitive reference strain included in Table 1, it is difficult for the reader to judge whether the criteria of an XDR strain are met. It would be helpful to include this information in Table 1.*

RESPONSE: We agree with your viewpoint, and we have added a sensitive reference strain ATCC 25922 to Table 1.

COMMENT: *Lines 160-163; Fig. 3: Is Tn2006 correctly depicted in Fig 3B? I think the interrupting double prime should be located between the genes encoding the ATPase and DEAD box helicase. Moreover, the color code in Fig. 3A is not referenced in the text. Please explain.*

RESPONSE: We apologize for the improper location of the interrupting double prime of Tn2006, and we have corrected in the Figure. The color codes in Fig. 3A have been added in Figure legends in Line 562-564.

COMMENT: *Lines 168-174; Fig. 5: Apparently, it escaped the attention of the authors that the isolated adeS allele carries an in-frame deletion, i.e. amino acids 32-53 are deleted, but all other amino acids are shared with the wild-type allele except for glycine residue at position 32. This is important for the interpretation of the activity of the encoded sensor kinase. Please correct the alignment in Fig. 5B accordingly. Is kinase or phosphatase activity of the kinase expected to be affected?*

How can constitutive activation of adeABC expression be explained by the deletion within AdeS. Is the response regulator known to autophosphorylate with acetyl-phosphate in vivo? The mutation in adeS detected in the current study needs to be re-evaluated by the authors.

RESPONSE: We apologize for the confusion generated for the inappropriate alignment, and we have corrected the alignment of AdeS amino acids in Figure 4B.

The sensor histidine kinase AdeS consists of two transmembrane N-terminal helices linked by a short extracellular sensor domain (residue 34-61), a HAMP domain (residue 84-138), a DHP (dimerization histidine phosphotransfer) domain (residue 146-204), and a C-terminal catalytic ATP (CA)-binding domain (residue 204-357) (Figure 2 listed below in ‘Response to Reviewers’) [6]. When AdeS receives an environment signal, it induces auto-phosphorylation through its histidine kinase domain and the phosphate group is then transferred to the response regulator. The phosphorylated regulator AdeR then controls expression of the AdeABC efflux pump by binding to a direct-repeat motif in the intercistronic spacer [7,8].

Previous publications have shown different mutation within AdeRS associated with overexpression of AdeABC efflux pump and tigecycline resistance [7,8]. The deletion part of AdeS in our study is located on the sensor domain, and will lead to the loss of its sensor domain and phosphorylation activity of kinase. The deletion within AdeS in our study like many mutation types of AdeS (GLY186, SER188, GLU121), will result in the prevention of auto-phosphorylation

of AdeS, and inhibit/slow the transfer of the phosphate group to AdeR, which can not be phosphorylated in other ways. Thus, AdeR may be binding loosely to the intercistronic spacer region leading to an overexpression of AdeABC.

In addition, the mutant AdeS in our study leads to the decrease of susceptibility to tigecycline, minocycline and levofloxacin, and substrates for the AdeABC efflux pump confirmed their roles in increasing pump expression. The description of the process of AdeS mutation affect AdeABC efflux pump have been added in the discussion part in Line 236-240.

Fig 2 Schematic view of histidine kinase AdeS domain organization

Reference :

6. Ouyang Z, Zheng F, Zhu L, Felix J, Wu D, Wu K, Gutsche I, Wu Y, Hwang PM, She J, Wen Y. 2021. Proteolysis and multimerization regulate signaling along the two-component regulatory system AdeRS. iScience 24:102476.

7. Chang TY, Huang BJ, Sun JR, Perng CL, Chan MC, Yu CP, Chiueh TS. 2016. AdeR protein regulates *adeABC* expression by binding to a direct-repeat motif in the intercistronic spacer. Microbiol Res 183:60-7.

8. Roy S, Junghare V, Dutta S, Hazra S, Basu S. 2022. Differential Binding of Carbapenems with the AdeABC Efflux Pump and Modulation of the Expression of AdeB Linked to Novel Mutations within Two-Component System AdeRS in Carbapenem-Resistant *Acinetobacter baumannii*. mSystems 7: e0021722.

COMMENT: *Lines 187-189: I think the MIC for Sulbactam decreased from 64 to 16 mg/ml.*

RESPONSE: Two numbers have been corrected (Line 185-186).

COMMENT: *Lines 271-276: Sulbactam is an irreversible inhibitor of β -lactamases but also known to display antibacterial activity in *A. baumannii* by targeting PBPs directly (i.e., PBP1a/b and PBP3), enzymes required for cell wall synthesis. Is it possible that a constitutively active*

AdeABC efflux pump may increase local concentration of Sulbactam in the exterior surrounding the cells, thereby leaving more of the drug that could subsequently inhibit the PBPs? Can this be tested?

RESPONSE: We thank the reviewer for pointing out this possible theory about how the overexpression of AdeABC efflux pump increase the susceptible of sulbactam. Actually, AdeBC efflux pump will pump the sulbactam from the inner of cell to the outside, and increase concentration of sulbactam in the exterior surrounding the cells [9]. However, sulbactam will enter the cell through channels on the outer membrane, like some porins [10, 11]. PBPs, as the antibacterial target of sulbactam, located on the peptidoglycan inside the outer membrane. Therefore, if just increase the external concentration of sulbactam, it probably won't affect the amount of sulbactam that binds to PBP (Fig 3 listed below).

In our study, the transcriptomic analysis of Ab-3557-*adeS*-mutation revealed the gene expression of Outer-membrane protein A(OmpA) was up-regulated, which may be also regulated by AdeRS. OmpA is β -barrel porin in *A. baumannii* and also play a role in antimicrobial resistance and virulence in *A. baumannii*. OmpA is mainly distributed on the outer membrane and the outer membrane vesicle (OMV) [11], so we speculated that the sulbactam susceptibility restoring might be mediated by the following mechanism: (a) Change the distribution of OmpA: increasing the distribution of OmpA on the outer membrane enlarges the channel of sulbactam into bacteria; (b) Decrease the secretion of OMV containing β -lactamase: because sulbactam is easily hydrolyzed by β -lactamses. The extracellular hydrolysis of sulbactam is reduced by decreasing the formation of OMV and the secretion of β -lactamases. The above hypothesis is currently being tested in a large number of experiments.

Fig 3. Schematic view of AdeABC efflux pump, porin and PBP.

Reference:

9. Xu C, Bilya SR, Xu W. 2019. adeABC efflux gene in *Acinetobacter baumannii*. New Microbes New Infect 30:100549.

10. Iyer R, Moussa SH, Durand-Reville TF, Tommasi R, Miller A. 2018. *Acinetobacter baumannii* OmpA Is a Selective Antibiotic Permeant Porin. ACS Infect Dis 4:373-381.

11. Skerniskyte J, Karazijaite E, Luciunaite A, Suziedeliene E. 2021. OmpA Protein-Deficient *Acinetobacter baumannii* Outer Membrane Vesicles Trigger Reduced Inflammatory Response. Pathogens 10:407.

Minor Comment:

COMMENT: *Is mentioning the isolate names in the abstract essential information? Please simplify the abstract.*

RESPONSE: We have simplified the abstract by deleting some unessential isolate name in the abstract.

COMMENT: *Lines 139-132: Can the distribution of the resistance genes be designated identical, if there was a duplication of oxa-23 in one of the isolates?*

RESPONSE: It was not accurate to be designated identical in the distribution of the resistance genes if there was difference in the copy number of bla_{oxa-23}. So we have rewritten the description in Line 125, the ‘identical’ word has been revised to ‘similar’.

COMMENT: *Line 194: Should read: “Strain Ab-3556 of the isolate pair 1 of the...”?*

RESPONSE: The sentence has been rewritten in Line 191.

COMMENT: *Line 198: “...reduced susceptibility to Sulbactam...”*

RESPONSE: The word has been corrected (Line 195).

COMMENT: *Line 228-229: “...novel antibiotic pair with tigecycline resistance and collateral...”*

RESPONSE: The word has been added in Line 226.

COMMENT: *Line 244-247: Please describe briefly what is known about the relationship between adeS mutations and tigecycline resistance as this is important information needed to evaluate the effects of the adeS in-frame deletion detected in the current study.*

RESPONSE: The description of the relationship between adeS mutation and tigecycline resistance have been added in the discussion part in Line 236-240.

COMMENT: *Line 267: First SUL, subsequently TGC? Please specify.*

RESPONSE: We have exchanged the order of sulbactam and tigecycline. *A. baumannii* strain easily acquired resistance to sulbactam when it was administered. However, based on the collateral sensitivity phenomenon in our study, the strain may restore sulbactam susceptibility if the regimen is switched to tigecycline. So the sequential combination of sulbactam and tigecycline may be an effective regimens used against XDR-AB.

COMMENT: *Lines 348-350: For convenience of the user, it would be advantageous to assign accession numbers to the corresponding strains. Currently, they are not in a particular order.*

RESPONSE: The genomes of strain Ab-3557 (CP104906- CP104907), YZM-0314 (CP104908-CP104910) and YZM-0406 (CP104911-CP104912) were submitted simultaneously, and strain Ab-3556 (CP104785-CP104787) was submitted separately, so the accession number of Ab-3556 was discontinuous with the other three strains.

COMMENT: *Line 551: β -lactamase*

RESPONSE: The word was corrected (Line 562).

April 13, 2023

Prof. Xianjun Wang
Affiliated Hangzhou First People's Hospital, Zhejiang University School of Medicine
Department of Clinical Laboratory
No.4 Xueshi Road
hangzhou
China

Re: Spectrum04594-22R1 (Alteration of *adeS* contributes to tigecycline resistance and collateral sensitivity to sulbactam in *Acinetobacter baumannii*)

Dear Prof. Xianjun Wang:

Your manuscript has been accepted, and I am forwarding it to the ASM Journals Department for publication. You will be notified when your proofs are ready to be viewed.

Sincerely,

Ayush Kumar
Editor, Microbiology Spectrum
